# Multi-Modality Adaptive Feature Fusion Graph Convolutional Network for Skeleton-Based Action Recognition

**DOI:** 10.3390/s23125414

**Published:** 2023-06-07

**Authors:** Haiping Zhang, Xinhao Zhang, Dongjin Yu, Liming Guan, Dongjing Wang, Fuxing Zhou, Wanjun Zhang

**Affiliations:** 1School of Computer Science, Hangzhou Dianzi University, Hangzhou 310005, China; zhanghp@hdu.edu.cn (H.Z.); yudj@hdu.edu.cn (D.Y.); dongjing.wang@hdu.edu.cn (D.W.); 2School of Information Engineering, Hangzhou Dianzi University, Hangzhou 310005, China; glm@hdu.edu.cn; 3School of Electronics and Information, Hangzhou Dianzi University, Hangzhou 310005, China; zxh8991@163.com (X.Z.); shiyi_zfx@hdu.edu.cn (F.Z.)

**Keywords:** action recognition, graph convolutional networks, feature fusion, attention mechanism

## Abstract

Graph convolutional networks are widely used in skeleton-based action recognition because of their good fitting ability to non-Euclidean data. While conventional multi-scale temporal convolution uses several fixed-size convolution kernels or dilation rates at each layer of the network, we argue that different layers and datasets require different receptive fields. We use multi-scale adaptive convolution kernels and dilation rates to optimize traditional multi-scale temporal convolution with a simple and effective self attention mechanism, allowing different network layers to adaptively select convolution kernels of different sizes and dilation rates instead of being fixed and unchanged. Besides, the effective receptive field of the simple residual connection is not large, and there is a great deal of redundancy in the deep residual network, which will lead to the loss of context when aggregating spatio-temporal information. This article introduces a feature fusion mechanism that replaces the residual connection between initial features and temporal module outputs, effectively solving the problems of context aggregation and initial feature fusion. We propose a multi-modality adaptive feature fusion framework (MMAFF) to simultaneously increase the receptive field in both spatial and temporal dimensions. Concretely, we input the features extracted by the spatial module into the adaptive temporal fusion module to simultaneously extract multi-scale skeleton features in both spatial and temporal parts. In addition, based on the current multi-stream approach, we use the limb stream to uniformly process correlated data from multiple modalities. Extensive experiments show that our model obtains competitive results with state-of-the-art methods on the NTU-RGB+D 60 and NTU-RGB+D 120 datasets.

## 1. Introduction

Actions play a particularly important role in human communication. These actions convey essential messages, such as feelings and root intentions, that help us to understand a person. Giving intelligent machines the same capabilities to understand human behavior is important for natural human–computer interaction and for many other practical applications that have attracted much attention in recent years. Today, modern sensor technology and algorithms for human position estimation make it much easier to access the 2D/3D human skeleton. Human skeleton data, which can be extracted from video images using pose estimation algorithms or captured directly using depth sensor devices, consist of time series of skeletal joints at multiple 2D or 3D coordinate positions. Compared with the traditional RGB video recognition method, the action recognition based on skeleton data has the advantage that the skeleton data can effectively reduce the influence of interference factors such as illumination changes, environmental background, and occlusion in the recognition process and are more adaptable to dynamic environments and complex backgrounds.

Early methods based on deep learning treated human joints as a series of individual features and organized them into characteristic sequences or pseudo-images. It is possible for RNN or CNN to predict the motion of a mark; however, these methods ignore the intrinsic correlation between joints and show that human topology is important information for the human skeleton. ST-GCN [1] introduces the GCN method and one-dimensional temporal convolution for the first time to extract motion features and graph structures to simulate the correlation between human joints. GCN-based methods have become increasingly popular, and many excellent works have emerged on this basis. AS-GCN [2] and 2s-AGCN [3] propose methods for adaptively learning the relationship between spatial joints from data. CTR-GCN [4] embeds three kinds of shared adjacency matrices divided according to the graph into the dynamic space in the channel dimension. However, most of these methods are biased towards modeling the spatial dimension, while ignoring the modeling of the temporal dimension. In the time dimension feature extraction work, the many existing works [1,2,3,5] relying only on fixed-size convolution kernels are far from sufficient. To model actions with different duration, recent works [4,6] introduce multi-scale temporal convolutions to improve and enhance ordinary temporal convolutions. These models use fixed-size convolution kernels at each layer of the network and use different dilation rates [7] to obtain larger receptive fields. Hence, some recent models [8,9] also use the previous multi-scale time modeling methods.

While multi-scale temporal convolution uses several fixed-size convolution kernels or dilation rates at each layer of the network, we argue this scheme is inflexible. Skeleton-based action recognition models are usually a network structure of stacked GCNs. GCN modules close to the network output tend to have larger receptive fields and can capture more contextual information. In addition, the receptive field of the GCN module close to the network input is relatively small [10]. It can be concluded that different layers have different effects on skeleton recognition. Therefore, for the learning of the temporal dimension, it is difficult to solve the problem of semantic detail features of skeletal actions by simply using several convolution kernels of fixed sizes or dilation rates in each layer of the network to achieve more effective modeling. Furthermore, most current models always extract spatial features first and connect the original input and the output of temporal features through residuals. Residual connections allow information to be directly transmitted to subsequent levels, thereby preserving the original features and avoiding feature disappearance layer by layer. However, in fact, the effective receptive field is not large, and there is a great deal of redundancy in the deep residual network, which will result in the loss of context when aggregating spatio-temporal information. In addition, approaches such as [3,11,12,13] use multi-stream networks to extract high-order features of skeletal data. This multi-stream approach is used by many advanced models. However, skeletons are considered as overall tree-like data structures. This means that, in some special actions, the entire input skeleton tree in each time series can be regarded as a whole to achieve good recognition results, just like many actions only need the participation of local body joints to be completed. For example, action categories such as “waving” and “victory” only involve hand joints, and action categories such as “walking” and “kicking” only involve leg joints [14]. There are some excellent body partitioning methods that split the skeleton tree into several separate part groups [15]. However, we believe that this is not sufficient, especially for actions that require the participation of most joints or body parts of the whole body, such as “running” and “high jump”.

To address some of the problems with the current model mentioned above, we introduce a new training framework named MMAFF that is based on skeleton-based behavior recognition. We propose a temporal modeling module with a multiscale adaptive attention feature fusion mechanism. First, extracting adaptive features of multiscale spatio-temporal topology for larger perceptual fields, instead of residual connections, we use an attention fusion mechanism that allows us to efficiently aggregate spatio-temporal scale features and solve the problem of context aggregation and initial feature integration. The temporal modeling module adapts the integration of topological features to help us to complete the simulation of actions. Specifically, the extracted spatial topology features of specific channels are input into our designed temporal adaptive feature fusion module. The temporal adaptive feature fusion module is divided into two parts. In the first part, we use multi-scale adaptive convolution kernels and dilation rates to optimize traditional multi-scale temporal convolution with a simple and effective self attention mechanism, allowing different network layers to adaptively select convolution kernels of different sizes and dilation rates instead of being fixed and unchanged. In addition, we have used an attention feature fusion mechanism that replaces the residual connection between the initial features and the output of the temporal module. We focus on the information of the initial features and the temporal dimension separately and then perform feature fusion, effectively fusing the initial features and high-dimensional temporal features, solving the problem of context aggregation and initial feature integration. Based on current multi-stream approaches, we propose a partial stream processing method called the limb stream. The limb stream integrates joint motion and bone motion modality data in the channel dimension. It uses fewer joints and network layers to train the joint motion stream and bone motion stream simultaneously, thus effectively reducing the number of training iterations and the number of parameters of the entire model. Due to the fact that most movements must be completed with the cooperation of the limbs, we consider the limbs as a whole that not only radiates some local detailed movements but also identifies joint movements that require the cooperation of multiple parts of the body. This allows for a more complete and centralized representation of the motion involving a subset of joints. The framework is shown in Figure 1. Our contributions mainly include the following points:(1)We optimize the traditional multi-scale temporal convolution to make it more adaptive and have the ability to fuse initial features so that a larger receptive field and local global context can be obtained.(2)We propose the limb stream, as a supplement to the traditional independent modality processing method, which can obtain finer features of limb joint group motion, enhance the recognition ability of the model, and perform final score fusion with joint stream and bone stream.(3)We conduct extensive experiments on NTU-RGB+D and-NTU RGB+D 120 to compare our proposed methods with the state-of-the-art models. Experimental results demonstrate the significant improvement of our methods.

The remaining chapters of this paper are organized as follows. Section 2 presents related work and recent progress. Section 3 details our proposed optimization. In Section 4, we compare our results with state-of-the-art methods and conduct ablation experiments. Section 5 summarizes the paper.

## 2. Related Work

### 2.1. Skeleton-Based Action Recognition

Skeleton-based action recognition is performed to classify actions based on inferred sequences of keypoints. Early deep learning methods used convolutional neural networks (CNNs) [16,17] or recurrent neural networks (RNNs) [18] to model the skeletons, but due to the inability to explicitly learn topology, recognition performance is limited by the skeleton structure. Recently, PoseC3D [19] modified CNN-based methods to collapse heatmaps into 3D volumes. This preserves the spatial and temporal features of the skeleton and greatly improves performance, but it has the disadvantage of high training overhead. In recent years, due to the natural composition of human joints and bones, using GCNs to extract high performance from skeletons has become a major trend in this field. Thus, the topology [1,3,4,6] of the human body is naturally flexible. ST-GCN [1] was the first work to use GCN as a feature extractor to model the skeleton by using a fixed diagram heuristic design. ST-GCN uses three joint partition methods to use data such as a spatio-temporal skeleton. The method also provides a Temporal Convolutional Network (TCN) for modeling temporal measurements. Based on this approach, MS-G3D [6] allows the direct propagation of information using edges at different temporal points, effectively improving topological spatio-temporal modeling. More recently, CTR-GCN [4] proposed an improved channel topology instead of shared topology, like the idea of dynamic graph convolution, which enables dynamic topology modeling in space. InfoGCN [8] develops an information rate-based framework for learning objectives. STF [10] provides a flexible framework for learning spatio-temporal gradients for skeleton-based action recognition. However, many advanced methods spend a great deal of effort on spatial feature extraction, ignoring the extraction of temporal features, and simply use the same multi-scale feature extraction module in each layer of the network. In fact, different layers require different receptive fields, so the temporal convolution kernel and dilation rate should also change with the number of layers of the network.

### 2.2. Attention Mechanism for Action Recognition

In the task of skeletal behavior recognition. Song et al. [20] initially offered a long-term method of short-term memory that models the differences between bony joints and spatio-temporal attention. Chiara et al. [21] proposed a new network of spatio-temporal transformers (ST-TR) that uses spatio-temporal transformers to model the expression of joint relations. Cheng et al. [22] added a downward pointing mechanism to the model as a means to improve regularization and effectively increase the accuracy of action recognition. Ye et al. [23] proposed a dynamic GCN method. They also proposed a due diligence approach that takes into account spatial relevance. Qiu et al. [24] proposed a method of spatio-temporal element transformers to capture the dependencies between different joints. In their recent work, Song et al. [25] proposed the Spatial Temporal Joint Attention module, which allows key joints to be found in a spatial and temporal sequence to better achieve efficient topology modeling. Zhou et al. [26] introduced a graph attention block based on Convolutional Block Attention Module (CBAM), which is used to calculate the semantic correlation between any two joints. TCA-GCN [27] used an MS-CAM attention fusion mechanism to solve the problem of the contextual aggregation of skeletal features. However, at present, the local and global context aggregation of skeletal features and the integration of initial features are still a major difficulty, and how to effectively integrate initial features into high-dimensional features needs further research.

### 2.3. Conventional Multi Stream Training

Some recent approaches [4,28] and earlier approaches [3,11] generate data of different modalities (bone and motion) based on the raw input skeleton data. Traditional methods use different modality data to train a particular architecture multiple times and then integrate the solutions. However, these traditional methods based on multiple streams will lead to a significant increase in the total number of parameters and are not very efficient. Song et al. [15] started research on a single modality pipeline very early on, integrating different modality streams in the early stage and centralizing them into the backbone network to achieve a single modality representation. However, before merging, each modality will be processed through several separate networks, which again invisibly increases the number of training parameters. Of course, we found that PSUMNet [14] proposed the idea of divisional stream to integrate the traditional stream state, as well as the training body, hand, and leg parts. However, it may be due to configuration reasons that our results in reproducing the PSUMNet [14] model are not very ideal. Hence, inspired by the partial stream and traditional stream state, we combined the two methods, and we performed a number of experiments to prove that our proposed limb stream state can achieve good results when applied to traditional stream state.

## 3. Method

In this section, we introduce specific method details for skeleton-based action recognition, as shown in Figure 2. Section 3.1 introduces the basic theory in this field. Section 3.2 introduces the components of MMAFF. We detail the specific architecture of the used SAGC module in Section 3.3. In Section 3.4, we describe the TAFF module in detail. In Section 3.5, we replace the commonly used multi-stream fusion method [4,29] with a multi-modality approach.

### 3.1. Preliminaries

In most skeleton behavior recognition tasks, the GCN-based method is usually used to construct the human skeleton structure as an undirected spatio-temporal graph G=(V,E), where *V* and *E* represent the sets of joints and bone edges, respectively. Consequently, $X\in R^{3\times T\times N}$ can be used to describe the temporal skeleton sequence, where *N* and *T* represent the number of joints and the size of the temporal window, respectively. According to the relationship between the joints and barycenter, the nodes are divided into three subgraphs to build the adjacency matrix. GCN’s operation with input feature map $X\in R^{C\times T\times N}$ is as follows:(1)$$F_{out}=\sum_{p\in P}A_{p}X\Theta_{p}$$
where $P=\{p_{id},p_{cf},p_{cp}\}$ denotes graph subsets, and $p_{id}$, $p_{cf}$, and $p_{cp}$ indicate identity, centrifugal, and centripetal joint subsets, respectively. $\Theta_{p}$ denotes the pointwise convolution operation. $A_{p}$ is the p-th channel shared adjacency matrix.

### 3.2. MMAFF

Next, we focus on the specific details of the MMAFF framework in the single-stream state, since different streams are shared under different network structures. Figure 2a is an overview of the network structure of the MMAFF single-stream state. The input initial skeleton data *X* are first processed by the Multi-Modality Adaptive Feature Fusion Framework (MMAFF). The processing results are transformed by global average grouping (GAP) and fully connected layers (FC) so that layer-by-layer predictions of individual streams can be obtained. MMAFF consists of multiple Spatio-Temporal Adaptive Fusion (STAF) blocks.

The specific structure of STAF is shown in Figure 2b. Each STAF block contains a Spatial Attention Graph Convolution module (SAGC) and a Temporal Adaptive Feature Fusion module (TAFF). The SAGC module in Figure 2c is used to dynamically extract the information from spatial dimensions, and the TAFF module in Figure 2d is used to adaptively extract temporal relations between joints and fuse multi-scale temporal features with initial features. Next, we elaborate on the details of each module.

### 3.3. SAGC Module

We first adopt [4] to construct a dynamic topology for spatial attention map convolution to dynamically model spatial attention maps. As shown in Figure 2c, we simultaneously feed the initial skeleton data into two parallel branches, each of which is processed by a 1 × 1 convolution and a temporal pooling block. Attention features are modeled by performing a subtraction operation on the pooled outputs of the two branches. This feature map is summed with the predefined adjacency matrix $A_{p}$ to obtain the final channel-wise topologies $A_{cwt}$, i.e., we use the attentional feature fusion mechanism to replace the ordinary residual connection to better aggregate the information of spatial and temporal scales. For details, see the TAFF module in Section 3.4.
(2)$$A_{cwt}=\beta Q\left(X_{in}\right)+A_{p}$$
where β is a learnable parameter, $A_{p}$ is the *p*-th channel shared topology, and *Q* is the topological relationship of the specific channel, defined as
(3)$$Q\left(X_{i}\right)=\sigma \left(TP\left(\phi \left(X_{in}\right)\right)-TP\left(\psi \left(X_{in}\right)\right)\right)$$
where σ, ϕ and ψ are 1 × 1 convolutions, TP is temporal pooling. After we obtain the channel-wise topologies $A_{cwt}$, we input the initial skeleton features into a 1 × 1 convolution and multiply the results with $A_{cwt}$ to aggregate the spatial dimension information as follows:(4)$$X_{out}=A_{cwt}⊗\left(\theta \left(X_{in}\right)\right)$$
where θ is a 1 × 1 convolution block. ⊗ is matrix multiplication operation.

### 3.4. TAFF Module

The multi-scale temporal adaptive feature fusion module consists of two parts: the TA module (temporal adaptive) and TFF module (temporal feature fusion). The first part can dynamically adjust the size of the convolution kernel and dilation rate at different network layers. As shown in Figure 2d, this module is improved on the basis of traditional multi-scale temporal convolution, which contains four branches. Each branch uses a 1 × 1 convolution to reduce the channel dimension. The two branches on the left are the core of the adaptive function. By introducing a simple attention mechanism, the size of the convolution kernel and the dilation rate can be dynamically adjusted. The convolution kernel size (ks) and dilation rate (dr) can be dynamically resized according to different dimensions of the output channel. Inspired by the attention mechanism [30], we use the following specific method formula:(5)$$t=abs(\frac{log(C_{l},2)+b}{gamma})$$
where $C_{l}$ is *l*-th network layer output channel dimension, and gamma and *b* are expressed as the parameters of the mapping function, set to 2 and 1, respectively. At layers 1–4 of the network, ks is 3, and at layers 5–10 of the network, ks is 5. Similarly, dr is 2 at layers 1–7 of the network, and at layers 8–10 of the network, dr is 3. Four branches of different scales obtain $X_{1}$ through the aggregation function. We did not introduce more branches, so there was almost no change in the parameter and computational complexity. Specific experiments can be found in Section 4.4.

For the second part, we use an attention feature fusion module to aggregate contextual information of different scales as well as different dimensions along the channel dimension. For the initial feature fusion problem in spatio-temporal modeling, we were inspired by TCA-GCN [27] model. Instead, we use the AFF module in [31] to fuse the features of different branches, and we use two input branches: one branch is *X* (the initial skeleton data), while the other branch is $X_{1}$ (multi-scale aggregated features). We focus on the information of the initial features and the temporal dimension separately and then perform feature fusion, effectively fusing the initial features and high-dimensional temporal features, solving the problem of context aggregation and initial feature integration and improving the effectiveness of modeling. As shown in Figure 2d, the above expression is specifically expressed as
(6)$$X^{\prime }=X⊗M\left(\cdot \right)+X_{1}⊗\left(1-M\left(\cdot \right)\right)$$
where *X* denotes the residual connection of the input, and $X_{1}$ is the concatenated output of multi-scale convolution. The specific formula for M(·) [31] is expressed as
(7)$$Sigmoid(L\left(X⊎X_{1}\right)⊕G\left(X⊎X_{1}\right))$$
where L(·) and G(·) are the local channel context and global channel context, respectively. Local context information is added to global context information within the attention module. Initial feature fusion is performed on the input features *X* and $X_{1}$. After sigmoid activation function, the output value is between 0 and 1. We hope to take the weighted average of *X* and $X_{1}$ and subtract this group of fusion weights by 1, which can be used as soft selection. Through training, the network can determine their respective weights.

### 3.5. Limb-Modality Generator

As mentioned before, we continued the multi-stream fusion scheme to train the joint and bone data separately, while integrating the joint-motion stream and bone-motion stream together, which we called the limb stream. Limb ($X_{L}$) includes all joints in the limb. For the NTU-RGB+D dataset, the number of joints for the limb stream is 22. Since most actions are performed by the limbs, the limbs can better reflect the characteristics of the motion information, and the global motion information can be propagated to a specific local joint group without losing the global motion information. Therefore, as shown in Figure 1, the training used data from three modalities: joint, bone, and limb.

Then, the subsequent integration of the solution was performed by performing weighted averaging on each predicted score of the modality to obtain the final classification. Changing the number of limb stream network layers may limit the total number of parameters used in the entire model. Replacing the motion stream with the limb stream can also reduce the training frequency and total training time.

Finally, we integrate joint motion and bone motion modality data in the channel dimension to generate $X\in R^{2C\times T\times N}$, which is the specific representation of the limb stream. We then use this as part of the input to the network. Concatenating the modality data helps to model the inter-modality relations in a more direct manner.

## 4. Experiments

### 4.1. Datasets

NTU-RGB+D 60. NTU-RGB+D 60 [32] is a large-scale human action recognition dataset containing 56,880 sample data, 60 actions, RGB video, a depth map sequence, 3D skeletal data, and infrared (IR) video for each sample. Each dataset is captured simultaneously by three Kinect V2 cameras. The RGB video has a resolution of 1920 × 1080, the depth map and IR video are both 512 × 424, and the 3D skeleton data contain 3D coordinates of 25 body joints per frame. Each sample contains one motion and is guaranteed to have up to two subjects simultaneously captured from different views by three Microsoft Kinect v2 cameras. The author of this dataset recommends two of her benchmarks: (1) cross-subject (X-sub): training data are taken from her 20 subjects, and test data are taken from her other 20 subjects; (2) cross-view (X view): training data are taken from camera views 2 and 3, and test data are taken from camera view 1.

NTU-RGB+D 120. NTU-RGB+D 120 [33] is currently the largest dataset with 3D joint annotations for human action recognition, which is a supplementary version of the previous version, covering all the previous data and adding an additional 60 categories. In total, 113,945 samples over 120 classes are performed by 106 volunteers, captured with three camera views. This dataset contains 32 setups, each representing a specific location and background. The authors of this dataset recommend two benchmarks: (1) cross-subject (X-sub): training data are obtained from 53 subjects, and test data are obtained from the other 53 subjects; (2) cross-setup (X-Setup): training data are sampled with even setup IDs, and test data are sampled with odd setup IDs.

### 4.2. Implementation Details

As shown in Figure 1, the input skeleton to each of the streams contains different numbers of joints. For NTU-RGB+D 60 and NTU-RGB+D 120 datasets, the joint and bone stream has an input skeleton with a total of 25 joints, while the limb stream has the input skeleton with a total of 22 joints. Within the MMAFF architecture, we use 10 STAFs for the joint and bone stream, and 8 STAFs (L1, L4-L10) to process the limb stream.

All experiments were conducted with the PyTorch deep learning framework. The Pytorch version was 1.7.1, and we set num-workers to 32. In addition, we used two NVIDIA A800 GPUs. The cross-entropy method is used as the loss function. Our models are trained with Stochastic Gradient Descent (SGD) with momentum (0.9). When training the model on the two datasets, we used the warmup strategy for the first 5 epochs. Considering the learning rate, we set this to 0.1 and used decays at 35 epochs and 55 epochs, ending the training at 80 epochs. For NTU-RGB+D 60 and NTU-RGB+D 120, we used the pre-processing [34], setting the batch size to 128. Joint, bone, and limb streams all used the same implementation configuration as above.

### 4.3. Comparison with State-of-the-Art Methods

Most recent state-of-the-art networks [4,22,29] adopt a four-way ensemble method, but we adopt the three-way ensemble method described in Section 3.5.

We compare our results with state-of-the-art networks on two datasets: NTU-RGB+D 60 [32] and NTU-RGB+D 120 [33]. Comparisons for each dataset are shown in Table 1 and Table 2. We report the results of three ensembles. Our model obtains state-of-the-art results on almost all benchmarks. Specifically, the best recognition effect was obtained with the NTU-RGB+D 120 dataset, indicating that our method has more advantages in identifying large datasets. Table 2 shows that our accuracy was improved by 0.2% and 0.3% compared to state-of-the-art models on the X-Sub and X-Set of NTU-RGB+D 120, respectively.

### 4.4. Ablation Study

In this section, we demonstrate the effectiveness of the proposed MMAFF. All experimental ablation studies are conducted on the NTU-RGB+D 60 and NTU-RGB+D 120 cross subject benchmark. As shown in Figure 2, we adopt [4] as our baseline network architecture with a total of 10 stacked GCN blocks.

TA Module. For the multi-scale temporal convolution module, we dynamically adjust the size of the convolution kernel and the size of the dilation rate according to the different dimensions of the output channels. Table 3 shows the improvement of classification accuracy by dynamically changing ks, and Table 4 shows the improvement of classification accuracy by dynamically changing dr. Msconv [4] is the multi-scale convolution. We validate the effect of the TA module. When we simultaneously change the size of the convolution kernel and dilation rate in Msconv, the overall effect of the TA module is shown in Table 5. We only show the effect on a single stream state, with an accuracy improvement of 0.2%. The computational and parameter quantities do not increase. If several branches were added, this would bring more improvements, and the corresponding inference time would also increase.

Contribution of each component. We scrutinize the contribution of each MMAFF component as shown in Table 6. We adopt [4] as our baseline network architecture. We observe that TA and TFF improve the baseline accuracy by 0.2% and 0.4% on NTU-RGB+D 60 cross subject, and the accuracy improved by 0.2% and 0.5% on NTU-RGB+D 120 cross subject. In the table, 4s represents the traditional fusion of four streams, while 3s represents the fusion of our joints, bones, and limb streams. Through experiments, we found that the total parameter quantity for 4s is 5.6 M, and the total training inference time is 72 h. Our model needs to be trained three times, with a total training inference time of 55.5 h, which is less than the baseline. However, the introduction of the AFF [31] module has resulted in an increase in the total number of parameters. These results demonstrated that the TAFF module guides our model to learn better representation for action classification.

Single-stream comparative ablation experiments. In order to verify the effect of our model based on the fusion of three stream states (joint, bone and limb), we compared the single stream state with the advanced method [4], as shown in Table 7. Our model outperforms the state-of-the-art methods by 0.4% and 0.5% for the joint stream and bone stream, respectively. We replace the two motion streams with a better-performing limb stream, which is 0.4% better than a single motion stream. We verify the effectiveness of the proposed multi modality skeleton-based action recognition model.

For the evaluation of the model, not only accuracy but also convergence speed should be considered. In order to comprehensively evaluate the model proposed in this article, we conducted experiments on the X-Sub and X-View benchmarks of the NTU-RGB+D 60 dataset. As our model was improved on the basis of CTR-GCN, we compared the convergence processes of CTR-GCN and MMAFF networks. Details are shown in Figure 3 and Figure 4.

It can be observed that as the number of iterations (epochs) increases, the recognition accuracy first rapidly increases and then slowly slows down. When the number of iterations (epochs) reaches a certain value, all curves can converge to the maximum value and then stabilize. Before 35 iterations, there may be some fluctuations in recognition accuracy, but the overall trend is upward. At the 36th iteration, the recognition accuracy of all curves will be significantly improved. Afterwards, the curve had almost no fluctuation amplitude.

Comparing the curves of two types of networks, the orange curve (MMAFF) performs almost the best on both X-Sub and X-View benchmarks, with small fluctuations, high recognition accuracy, and fast convergence speed.

## 5. Conclusions

In this paper, we propose a multi-modality adaptive feature fusion framework (MMAFF) to increase the receptive field of the model in spatial and temporal dimensions. Firstly, we propose a TAFF module that includes a TA module and TFF module, which can dynamically adjust the size of convolution kernel and dilation rate at different network layers and aggregate multi-scale context information along channel dimensions. Then, we introduce the limb stream; as a supplement to the traditional independent modal processing method, the limb stream enables richer and more dedicated representations for actions involving a subset of joints. Extensive experiments show that our model has advanced performance on different types of datasets.

In future research, we plan to extract features from both temporal and spatial graph dimensions simultaneously, rather than extracting spatial information first and then temporal information, in order to reduce feature redundancy. We will consider introducing language text models in the future to improve the performance of action recognition and reduce computational complexity.

## Figures and Tables

**Figure 1 sensors-23-05414-f001:**
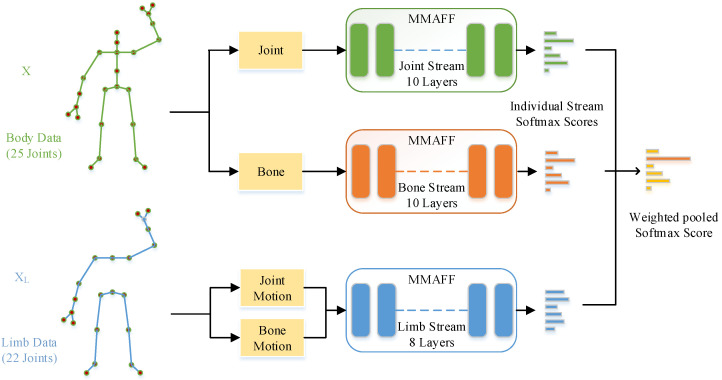
We used data from the three modalities of joints, bones, and limbs for training; then, we input the skeleton data of different modalities into the network framework, output the independent stream state softmax score, and finally performed score fusion on the independent output to obtain the final classification score—see Figure 2 for architectural details of MMAFF.

**Figure 2 sensors-23-05414-f002:**
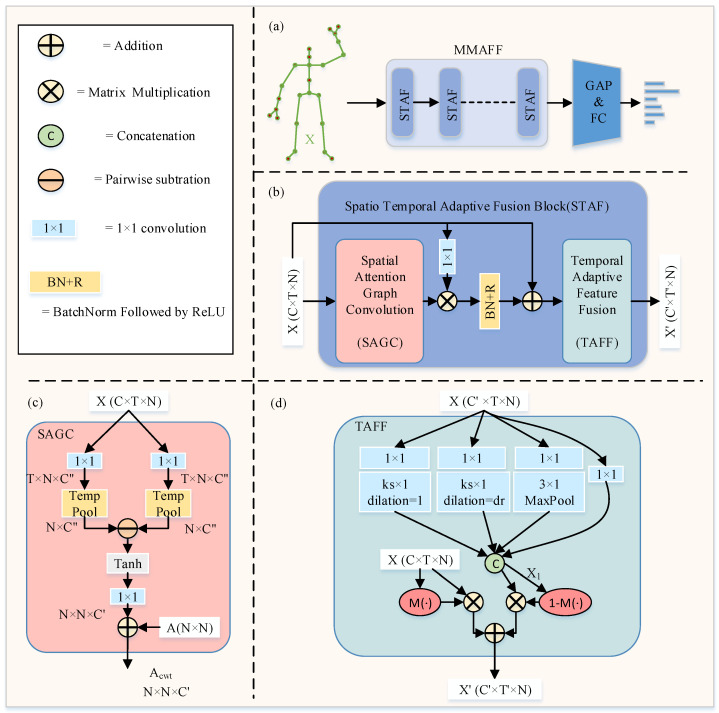
(**a**–**d**) MMAFF is a specific structure diagram of a single-stream state, and STAF is a spatio-temporal adaptive fusion module. Each STAF module consists of an SAGC and a TAFF module. The ks in (**d**) represent the size of the convolutional kernel, while dr represents the size of the dilation rate. Their values can vary with the number of network layer channels. *X* is the initial skeleton feature, $X_{1}$ is the temporal dimension feature. *X* and $X_{1}$ are fused to output $X^{\prime }$ through the self attention mechanism.

**Figure 3 sensors-23-05414-f003:**
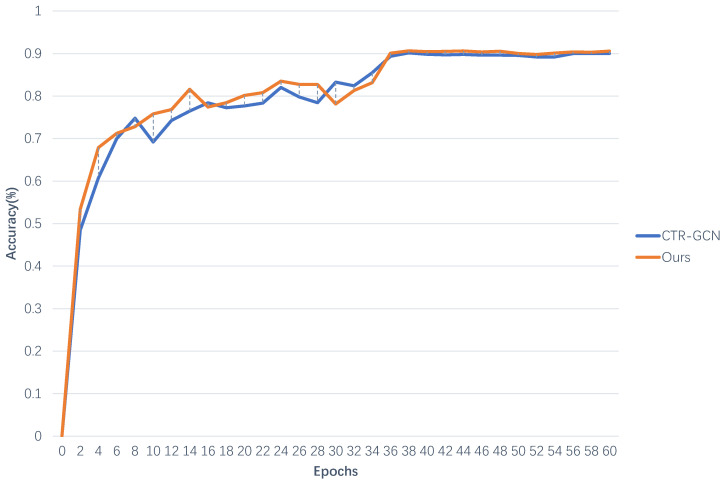
Convergence process of bone stream for two types of networks on X-Sub benchmark.

**Figure 4 sensors-23-05414-f004:**
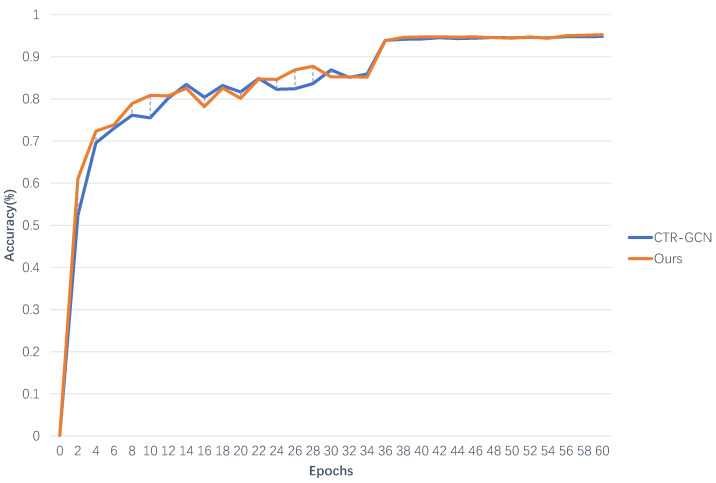
Convergence process of joint stream for two types of networks on X-View benchmark.

**Table 1 sensors-23-05414-t001:** Comparison of top-1 accuracy (%) with state-of-the-art methods for NTU-RGB+D 60 dataset.

Methods	X-Sub (%)	X-View (%)	Years
ST-GCN [1]	81.5	88.3	2018
AS-GCN [2]	86.8	94.2	2019
2s-AGCN [3]	88.5	95.1	2019
DGNN [5]	89.9	96.1	2019
SGN [34]	89.0	94.5	2020
Shift-GCN [11]	90.7	96.5	2020
DC-GCN + ADG [22]	90.8	96.6	2020
DDGCN [35]	91.1	97.1	2020
MS-G3D [6]	91.5	96.2	2020
MST-GCN [36]	91.5	96.6	2021
EfficientGCN-B4 [25]	91.7	95.7	2021
CTR-GCN [4]	92.4	96.8	2021
STF [10]	92.5	96.9	2022
ST-GCN++ [37]	92.6	97.4	2022
Info-GCN [8]	92.7	96.9	2022
TCA-GCN [27]	92.8	97.0	2022
PSUMNet [14]	92.9	96.7	2022
FR Head [38]	92.8	96.8	2023
STF-Net [39]	91.1	96.5	2023
RSA-Net [40]	91.8	96.8	2023
Our approach (3 ensemble)	93.1	96.9	2023

**Table 2 sensors-23-05414-t002:** Comparisons of the top-1 accuracy (%) against state-of-the-art methods on the NTU-RGB+D 120.

Methods	X-Sub (%)	X-Set (%)	Years
ST-GCN [1]	70.7	73.2	2018
2s-AGCN [3]	82.5	84.2	2019
SGN [34]	79.2	81.5	2020
Shift-GCN [11]	85.9	87.6	2020
DC-GCN + ADG [22]	86.5	88.1	2020
MS-G3D [6]	86.9	88.4	2020
MST-GCN [36]	87.5	88.8	2021
EfficientGCN-B4 [25]	88.3	89.1	2021
CTR-GCN [4]	88.9	90.6	2021
STF [10]	88.9	89.9	2022
ST-GCN++ [37]	88.6	90.8	2022
Info-GCN [8]	89.4	90.7	2022
TCA-GCN [27]	89.4	90.8	2022
PSUMNet [14]	89.4	90.6	2022
FR Head [38]	89.5	90.9	2023
STF-Net [39]	86.5	88.2	2023
RSA-Net [40]	88.4	89.7	2023
Our approach (Joint)	85.2	86.8	2023
Our approach (Bone)	86.3	88.0	2023
Our approach (Limb)	81.5	83.1	2023
Our approach (Joint + Bone)	89.1	90.3	2023
Our approach (3 ensemble)	89.7	91.2	2023

**Table 3 sensors-23-05414-t003:** The effectiveness of the dynamic kernel size ks in our TA module, compared with the traditional Msconv.

Methods	Output Channel			Acc (%)
	64	128	256	
Msconv	5	5	5	89.8
kernel size	3	5	5	89.9

**Table 4 sensors-23-05414-t004:** The effectiveness of the dynamic dilation rate dr in our TA module, compared with the traditional Msconv.

Methods	Output Channel			Acc (%)
	64	128	256	
Msconv	2	2	2	89.8
dilation rate	2	2	3	90.0

**Table 5 sensors-23-05414-t005:** Comparison between TA module and traditional Msconv.

Methods	Params (M)	Acc (%)
Msconv (Joint)	1.4	89.8
TA (Joint)	1.4	90.0

**Table 6 sensors-23-05414-t006:** Comparison of classification accuracy when applying each module of our proposed MMAFF to the baseline.

Methods	Params (M)	NTU60	NTU120	Time
Baseline (4s)	5.6	92.4	88.9	72 (h)
w TA (3s)	5.6	92.6 (+0.2)	89.1 (+0.2)	54 (h)
w TFF (3s)	7.9	92.8 (+0.4)	89.4 (+0.5)	54 (h)
w TA, TFF (3s)	7.9	93.1 (+0.7)	89.7 (+0.8)	55.5 (h)

**Table 7 sensors-23-05414-t007:** Performance comparison of our model with traditional standalone streams on different data streams.

Stream	CTR-GCN	Ours
Joint	89.8	90.2 (+0.4)
Bone	90.2	90.7 (+0.5)
Limb	-	87.8
Joint motion	87.4	-
Bone motion	86.9	-

## Data Availability

The code can be accessed at https://github.com/momo222333/MMAFF.git (accessed on 25 April 2023).

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
