# Peer review of "Multi-Modality Adaptive Feature Fusion Graph Convolutional Network for Skeleton-Based Action Recognition"

_sensors, 2023, doi:10.3390/s23125414_

Round 1
Reviewer 1 Report
After careful review, I have the following feedback:
From the proposed method framework, it seems that there is no significant difference from most of the multi-stream integration-based methods. Methods based on integration usually improve performance. You may refer to the following paper for more information: [Action Recognition Based on the Fusion of Graph Convolutional Networks with High Order Features].
What are the "ks" and "dr" means in Tables 3 and 4? From Tables 3 and 4, there does not seem to be a significant improvement. To demonstrate the efficiency of the proposed modules more clearly, you should compare the parameters in Tables 3, 4, and 5.
In Table 5, the training time and inference time should be compared to improve the fairness of the comparison.
In lines 58-61, you pointed out some issues with the current methods, for example, " most current models always extracting spatial features first ..." . However, you did not explain how you solved this problem and how your method differs from others.
In line 61, a punctuation mark is redundant. Additionally, please pay attention to the capitalization of letters.
There is an error in line 84. Please check this carefully.
The English expression is good, but there are some minor errors. Please check carefully and correct it.
Reviewer 2 Report
In this study, the authors proposed a Multi-modality adaptive Feature Fusion Graph Convolutional Network for Skeleton-Based Action Recognition. Although the performance seems promising, some major points should be addressed as follows:
1. The manuscript describes a framework for skeleton-based motion recognition that aims to address some common issues with conventional methods. While the proposed framework may have some new ideas, it is important to determine whether the approach is truly novel or builds upon existing work in the field. If the framework is not substantially different from previous approaches, the paper may not be considered significant enough to warrant publication.
2. It is important to ensure that the proposed approach is evaluated using appropriate metrics and datasets. It is essential to include comparisons with state-of-the-art methods in the field, and the evaluation should be comprehensive and rigorous.
3. The manuscript should provide sufficient details about the implementation of the proposed approach to allow other researchers to reproduce the results. Providing open-source code and pre-trained models would further increase the reproducibility of the work.
4. Uncertainties of models should be reported.
5. When comparing the performance results among methods/models, the authors should perform some statistical tests to see significant differences.
6. More discussions should be added.
7. Overall, English writing should be improved.
Overall, English writing should be improved.
Reviewer 3 Report
The paper proposed a multi-modality adaptive feature framework to simultaneously increase the receptive field in both spatial and temporal dimensions for action recognition.
However, the following concerns are raised:
1. Nearly half of the abstract is used to introduce the problem, while not enough info about the proposed method is provided.
2. Suggest providing more details in section 3 method, such as parameters analysis, network structure analysis, etc. Also, more implementation details in section 4.2 should be provided for readers to repeat the experiments.
3. In abstract, it says” we propose a multi-modality adaptive feature framework (MMAFF)” line 7~8. In section 3.1, it says” Multimodal Adaptive Feature Fusion Framework (MMAFF)” in line 184~185. What does the last “F” mean, fusion or framework? What does “MM” mean, multi-modality or multimodal?
4. Experiment results conducted on NTU 120(NTU 60 is included) are provided. Since the paper’s contributions focus on the methods, more experiments on different datasets are encouraged.
The paper is easy to follow for me.
Round 2
Reviewer 2 Report
My previous comments have been addressed.
My previous comments have been addressed.